# Synthesis and Molecular Docking of Some Novel 3-Thiazolyl-Coumarins as Inhibitors of VEGFR-2 Kinase

**DOI:** 10.3390/molecules28020689

**Published:** 2023-01-10

**Authors:** Tariq Z. Abolibda, Maher Fathalla, Basant Farag, Magdi E. A. Zaki, Sobhi M. Gomha

**Affiliations:** 1Department of Chemistry, Faculty of Science, Islamic University of Madinah, Madinah 42351, Saudi Arabia; 2Department of Chemistry, Faculty of Science, Zagazig University, Zagazig 44519, Egypt; 3Department of Chemistry, Faculty of Science, Imam Mohammad Ibn Saud Islamic University (IMSIU), Riyadh 11623, Saudi Arabia; 4Department of Chemistry, Faculty of Science, Cairo University, Cairo 12613, Egypt

**Keywords:** acetylcoumarin, hydrazonoyl halides, thiazoles, molecular docking, VEGFR-2

## Abstract

One crucial strategy for the treatment of breast cancer involves focusing on the Vascular Endothelial Growth Factor Receptor (VEGFR-2) signaling system. Consequently, the development of new (VEGFR-2) inhibitors is of the utmost importance. In this study, novel 3-thiazolhydrazinylcoumarins were designed and synthesized via the reaction of phenylazoacetylcoumarin with various hydrazonoyl halides and α-bromoketones. By using elemental and spectral analysis data (IR, ^1^H-NMR, ^13^C-NMR, and Mass), the ascribed structures for all newly synthesized compounds were clarified, and the mechanisms underlying their formation were delineated. The molecular docking studies of the resulting 6-(phenyldiazenyl)-2*H*-chromen-2-one (**3**, **6a–e**, **10a–c** and **12a–c**) derivatives were assessed against VEGFR-2 and demonstrated comparable activities to that of Sorafenib (approved medicine) with compounds **6d** and **6b** showing the highest binding scores (−9.900 and −9.819 kcal/mol, respectively). The cytotoxicity of the most active thiazole derivatives **6d**, **6b**, **6c**, **10c** and **10a** were investigated for their human breast cancer (MCF-7) cell line and normal cell line LLC-Mk2 using MTT assay and Sorafenib as the reference drug. The results revealed that compounds **6d** and **6b** exhibited greater anticancer activities (IC_50_ = 10.5 ± 0.71 and 11.2 ± 0.80 μM, respectively) than the Sorafenib reference drug (IC_50_ = 5.10 ± 0.49 μM). Therefore, the present study demonstrated that thiazolyl coumarins are potential (VEGFR-2) inhibitors and pave the way for the synthesis of additional libraries based on the reported scaffold, which could eventually lead to the development of efficient treatment for breast cancer.

## 1. Introduction

According to statistics, among all cancers affecting women, breast cancer accounts for 33.1 percent, making it the most prevalent disease of either gender [1]. Breast cancer early detection programs were a valuable resource for the tens of thousands of women who were diagnosed with cervical cancer or malignant or premalignant breast cancer [2]. The percentages of incidence and fatality, however, remain at historically high levels [3,4]. Invasive breast cancer has been used to express a number of angiogenic factors at all tumor stages [5]. Additionally, vascular endothelial growth factor receptor-2 (VEGFR-2) has been discovered to be significantly expressed in both primary and metastatic invasive breast carcinomas, suggesting a role for the VEGF signaling pathway in the regulation of breast tumor angiogenesis [6]. The C-terminal and N-terminal lobes each contributed residues to the region of the VEGFR-2 of protein kinases, which actively bind adenosine triphosphate (ATP), which is placed in the gap between the two lobes [7]. At the lobe, the C-terminal is an activation loop that has a conserved aspartate-phenylalanine-glycine (DFG) motif at the start of it [8]. Type I through III inhibitors are the three classes of VEGFR-2 inhibitors. Type II inhibitors maintain the DFG motif-containing DFG-out conformation of inactive VEGFR2 kinase; Type II inhibitors create a hydrophobic allosteric pocket next to the ATP-binding site. Improved kinase selectivity and high cellular potency are just two benefits of type II inhibitors [9]. Through the suppression of the Ras/MAPK pathway, VEGFR-2 inhibitors also delayed the development of selective estrogen receptor modulator (SERM) resistance in breast cancer [10]. An important field of study in the fight against cancer is the discovery and creation of novel anticancer drugs with high efficacy and low toxicity. According to reports in the literature, compounds comprising coumarins, thiazoles, or thiazolylcoumarins have drawn a lot of attention from drug research due to their potential anticancer action with good IC_50_ [11,12] (Figure 1).

Coumarin is a versatile molecule that serves as the pharmacological and biological building block for a wide range of naturally occurring chemicals [13,14,15]. It has been regarded as an intriguing framework for the development of anticancer drugs [16,17,18,19]. Furthermore, recent research revealed that a variety of coumarin compounds, both natural and synthetic, have antiproliferative properties via inhibiting VEGFR-2-mediated signaling pathways (Figure 2) [20,21,22,23]. On the other hand, thiazoles are considered to be important chemical synthons found in a variety of pharmacologically active compounds [24]. They possess a wide range of biological activities as anticancer, antimicrobial and anti-inflammatory agents [25,26,27]. Some thiazole derivatives were reported as type II VEGFR-2 inhibitors with similar activity compared with the Sorafenib reference drug (Figure 2) [28,29,30,31,32,33,34,35,36,37,38].

In light of our previous work on the synthesis of novel antitumor heterocycles [39,40,41,42,43,44,45,46] and with consideration of the aforementioned results, a new sort of VEGFR-2 inhibitors has been developed as prospective anti-breast cancer agents by hybridizing the coumarin and 1,3-thiazole moieties, which have been found to inhibit kinases and have antiproliferative properties. In this study, we developed and synthesized new 3-thiazolhydrazinylcoumarins in an effort to enhance the target compounds’ synergistic pharmacological significance and assess their anti-breast cancer activity targeting VEGFR-2. Finally, in order to occupy the hydrophobic back pocket of VEGFR-2, a side phenyl ring was maintained, either mono or di, substituted with a wide variety consisting of hydrophobic groups, such as chloro, methyl, and phenyl. The molecular docking studies of these compounds were performed to confirm their ability to satisfy the pharmacophoric features. Moreover, it also determines the binding mode interaction that occurred with the desired VEGFR-2 inhibition.

## 2. Results and Discussion

### 2.1. Chemistry

Our research aims to synthesize a new series of bioactive thiazole derivatives, and this may be performed by synthesizing the starting derivative 2-(1-(2-oxo-6-(phenyldiazenyl)-2*H*-chromen-3-yl)ethylidene)hydrazine-1-carbothioamide (**3**) via the reaction of 3-acetyl-6-(phenyldiazenyl)-2*H*-chromen-2-one (**1**) [47] and hydrazinecarbothioamide **2** in EtOH in the presence of a catalytic amount of HCl under reflux as depicted in Figure 1. Element and spectral data techniques (IR, ^1^H-NMR, mass) were used to determine structure **3** (see Section 3 experimental section). 

The reaction of compound **3** with hydrazonoyl chlorides **4a–e** [48] in EtOH containing Et_3_N yielded the thiazole derivatives **6a–e** via cyclization with the removal of the H_2_O molecule from intermediate **5** (Figure 1). The structure of product **6** was proved by spectral (IR, mass, ^1^H-NMR, ^13^C-NMR) and elemental data. The ^1^H-NMR spectrum of product **6a** showed a singlet signal at *δ =* 10.36 ppm assigned to the -NH proton, in addition to the usual signals of the fourteen aromatic protons and the two CH_3_ group protons. Furthermore, the IR spectrum showed two stretching bands at υ = 1725 and 3347 cm^−1^ due to the C=O and the NH groups. Its ^13^C-NMR spectrum showed *δ =* 8.4, 14.7 (2CH_3_), 116.3–162.1 (20Ar-C and C=N) and 163.5 (C=O) ppm. Moreover, the mass spectra of all derivatives of compound **6** showed molecular ion peaks at the right molecular weight for the corresponding molecules (see the Section 3 experimental section, Appendix A). 

It is envisioned that the nucleophilic attack of the thiol group of compound **3** at the electron-deficient carbon of the hydrazone group of compound **4** creates an intermediate **5**, which would then undergo a dehydrative cyclization to produce the final products of compound **6**.

Alternative synthetic techniques might be used to create authentic samples of **6a**. Product **6a** was obtained as a result of the reaction of phenyl diazonium chloride and 3-(1-(2-(4-methyl-5-(phenyldiazenyl)thiazol-2-yl)hydrazineylidene)ethyl)-2*H*-chromen-2-one (7) [49] in pyridine (Figure 1).

On the other hand, the reaction of compound **3** with ethyl 2-chloro-2-(2-arylhydrazineylidene)acetate **8** [50] in refluxing EtOH containing TEA as a basic catalyst afforded products **10** (Figure 1). The structures of product **10** were confirmed based on spectral (IR, mass, ^1^H-NMR, ^13^C-NMR) and elemental data. The ^1^H-NMR spectrum of **10a** revealed the expected signals at *δ* = 2.26, 2.27 (2s, 2CH_3_), 7.02–7.92 (m, 13 Ar-H), 10.54, 10.79 and (2 br s, 2 NH) ppm. Also, the ^13^C-NMR spectrum showed the expected signals at *δ* = 8.9, 14.2 (2CH_3_), 117.0–156.7 (19 Ar-C and C=N), and 163.0, 171.3 (2C=O) ppm. In addition, its IR spectrum showed the expected characteristic stretching bands at υ = 3413, 3289 (2NH) and 1724, 1692 (2C=O) cm^−1^ (see Section 3 experimental section, Appendix A).

Furthermore, the reaction of compound **3** with α-bromoketones was used to investigate the potential of compound **3** as a building block for the production of another series of predicted physiologically active thiazoles. Thus, the reaction of compound **3** with substituted phenacyl bromides **11a–c** in EtOH afforded thiazoles **12a–c** (Figure 2). The ^1^H-NMR spectrum of product **12a**, considered an example of product **12**, revealed the predicted signals assigned for the CH_3_, fourteen aromatic protons, and NH at *δ* = 2.31, 6.94–8.42, and 11.32 ppm, respectively. In addition, its ^13^C-NMR spectrum showed *δ* 10.1(CH_3_), 106.1–159.6 (20Ar-C and C=N) and 164.5 (C=O) ppm. The mass spectra of compounds **12a–c** exhibited a peak that matched their molecular ions in each case. Infrared spectra showed four bands, each corresponding to the carbonyl and NH groups, at υ = 1722 and 3327 cm^−1^.

### 2.2. Molecular Docking

The molecular docking was performed using MOE (Molecular Operating Environment) software version 2015.10 and was confirmed by re-docking the native ligand Sorafenib in the VEGFR-2 active regions giving a docking score of −9.284 kcal/mol. The modes of Sorafenib’s interactions with VEGFR-2’s active site residues are demonstrated in Figure 3. It is evident that the urea linker of Sorafenib plays a significant role in its binding ability for the enzyme VEGFR2 as it fits into the enzyme’s allosteric site to form four significant hydrogen bonds with four essential residues (one *H*-bond acceptor between chloro atom with Ile1025, one *H*-bond acceptor between the oxygen of urea moiety and Asp1046, and finally two *H*-bonds donors between two NH of urea side chain with Glu885). The hydrophobic 4-chloro-3-trifluoromethylphenyl is directed toward the hydrophobic back pocket by the urea linker’s binding mechanism. In addition, one *H*-bond acceptor between *N* atom of pyridine with essential amino acid Cys919 (inserted in Figure 3 as a 2D view). 

Molecular docking studies demonstrated that the synthesized coumarin-based derivatives (**3**, **6a–e**, **10a–c** and **12a–c**) interact with the VEGFR-2 enzyme active site in similar ways to those of Sorafenib with binding energy values between −7.723 to −9.900 kcal/mol. The energy scores and receptor interactions of type II VEGFR-2 inhibitor for the synthesized compounds compared to the native ligand (Sorafenib) are summarized in Table 1 and Table 2. 

For example, compound **6a** in type II VEGFR-2 inhibitor showed two *H*-bond; one *H*-bond donor between the carbon phenyl of 2*H*-chromen-2-one skeleton with Asp814, and another *H*-bond acceptor between the nitrogen of thiazole ring with Asp1046, respectively, but compounds **6a**, **b** in type I VEGFR-2 inhibitor showed *H*-bond acceptor between the nitrogen atom with Lys868 (inserted in Figure 4 and Figure 5 as a 2D view). 

Compound **6b** in type II VEGFR-2 inhibitor form two *H*-bond acceptors; one between the nitrogen atom of thiazole moiety and Asp1046 and another with Arg1027. In addition, π-*H* bond interaction with Lys868. Compound **6c** forms an *H*-bond acceptor with the Lys868. As depicted in Figure 4, compounds **6d** and **6e** form one *H*-bond acceptor between the nitrogen atom of the azo moiety and the side chain of Asp1046. Moreover, there was π-*H* interaction between the phenyl scaffold and Lys868. Moreover, compound **6e** showed an additional *H*- bond donor between the sulfur atom of the thiazole ring and Asp1046. 

On the other hand, compounds **10a** and **b** form an *H*-bond acceptor with the Asp1046 via the nitrogen atom of the thiazole ring. Moreover, compound **10b** in type II VEGFR-2 inhibitor showed an additional *H*-bond acceptor between the carbonyl group of the thiazole ring and Phe1047. However, this compound in type I VEGFR-2 inhibitor showed two *H*-bond donors between the nitrogen atom and the sulfur atom of the thiazole ring with Asp814 (inserted in Figure 5 as a 2D view). Compound **10c** exhibits two *H*-bond acceptors between the two oxygen atoms of the nitro group with the side chain of Cys919 and Gly922, respectively. Moreover, there was one π-*H* interaction between the thiazole ring and Lys868 (inserted in Figure 4 and Figure 5 as a 3D view). 

Compounds **12a** and **b** form two *H*-bond donors with Asp814 via NH and the sulfur atom of thiazole moiety. Moreover, compound **12a** exhibits an additional *π-H* interaction between the thiazole ring and Arg1027. Compound **12b** shows an additional *H*-bond acceptor between the nitrogen atom of the azo group and Asp1046. In addition, two π-*H* interactions, one between the phenyl ring and Lys868, and another π-*H* interaction between the phenyl of 2*H*-chromen-2-one skeleton and Leu889. These latter π-*H* interactions of **12b** are present in compound **3** in addition to two *H*-bond donors with Asp814 and one *H*-bond acceptor with Asp1046. Compound **12c** in type II VEGFR-2 inhibitor showed three *H*-bonds; one *H*-bond donor between the sulfur atom of the thiazole ring and Ser884, and two *H*-bond acceptors between the oxygen atom of the nitro group between Lys868 with Leu1049, respectively. In addition, three π-*H* interactions; one between the phenyl ring with Lys868, another π-*H* interaction between the phenyl of 2*H*-chromen-2-one skeleton with Leu889, and finally, π-*H* interaction between the phenyl of the 4-nitrophenyl skeleton with Glu885. However, this compound in type I VEGFR-2 inhibitor exhibits two *H*-bond donors between the nitrogen atom and the sulfur atom of the thiazole ring with the side chain of Asp814 (inserted in Figure 5 as a 3D view). The results implied that the ligands under investigation occupy similar positions and orientations within Sorafenib’s hypothesized binding sites.

### 2.3. Cytotoxic Potential

Using the MTT test and Sorafenib as a reference medication, the cytotoxicity of the most active synthesized thiazole derivatives **6b**, **6c**, **6d**, **10a**, and **10c** for their human breast cancer (MCF-7) cell line and normal cell line LLC-Mk2 was examined. Afterward, the determination of the tested sample concentrations that were sufficient to kill 50 percent of the cell population (IC_50_) was done by using the cytotoxicity results in plotting a dose-response curve.

In addition, cytotoxic activities were reported as the average IC_50_ from three separate tests. Table 3 and Figure 6 demonstrate that most of the evaluated compounds had very varied activity when compared to the reference drug.

Examination of the SAR leads to the following conclusions:

The 1,3-thiazoles **6d** and **6b** (IC_50_ = 10.5 ± 0.71 and 11.2 ± 0.80 μM, respectively) demonstrated promising anticancer activity against MCF-7 and outperformed the reference drug (IC_50_ = 5.10 ± 0.69 μM).

For 1,3-thiazoles **6**: Substitution of the phenyl group at position 5 in the 1,3-thiazole ring with Cl atom (electron-withdrawing atom) enhances the anticancer activity (**6d > 6b**, **6c**).

It was observed that the chloro **6d** derivative is more active than its methyl **6b** counterpart, which may be owing to the influence of substituent lipophilicity and the atomic size; hence, **6d** is the most potent derivative. Thus, the orientation of compound **6d** in the pocket allowed effective hydrophobic interactions as compared to compound **6b** (see the 3D models of both compounds, Figure 3). Additionally, the electron-withdrawing effect of the chloro substituent of **6d** has a greater stimulatory effect on activity than the electron-donating methyl substituent of **6b**. On the other hand, 2,4-dichloro derivative **6e** was more active than the unsubstituted one **6a**.

For 1,3-thiazolones **10**: The introduction of an electron-withdrawing group (e.g., NO_2_) at the para-position of the phenyl group at position 5 in the 1,3-thiazole ring enhances the antitumor activity (**10c** > **10a**). 

The introduction of the p-substitution of the electron-withdrawing group at derivative **12** has a similar activity enhancement effect. Thus, compound **12c** was the most reactive in this series, and the unsubstituted derivative **12a** exhibited the lowest activity. In addition, derivatives with thiazole moiety exhibited higher activity as compared to compound **3**, thus highlighting the positive effect of the thiazole ring on the activity of the reported compounds.

The 1,3-thiazole derivatives **6** have higher anticancer activity towards MCF-7 cell lines as compared to thiazolone derivatives **10**. 

The cytotoxic activity of the Sorafenib standard drug and most active compounds **6b**, **6c**, **6d**, **10a** and **10c** were also estimated on LLC-Mk2 (rhesus monkey kidney epithelial normal cells). The outcomes of these measurements demonstrated the non-toxic effect of the tested derivatives because their CC_50_ toward normal cell lines is higher than 100 μM, as shown in Table 3.

## 3. Experimental Section

See the Appendix A.

Synthesis of *2-(1-(2-oxo-6-(phenyldiazenyl)-2H-chromen-3-yl)ethylidene) hydrazine-1-carbothioamide* (**3**). 

A mixture of 3-acetyl-6-(phenyldiazenyl)-2*H*-chromen-2-one (**1**) (10 mmol, 2.92 g) and hydrazinecarbothioamide (**2**) (10 mmol, 0.91 g) in EtOH (40 mL) containing a few drops of concentrated HCl was heated under reflux for 4 h. The formed precipitate was recrystallized from EtOH to give product **3** as a yellowish-white solid in 73% yield; m.p. 177–179 °C; ^1^H-NMR (DMSO-*d_6_*) *δ*: 2.41 (s, 3H, CH_3_), 7.15–8.32 (m, 11H, Ar-H and NH_2_), 10.39 (s, br, 1H, NH) ppm; IR (KBr) *ν* cm^−1^: 3419, 3372, 3284 (NH_2_ and NH), 1725 (C=O), 1606 (C=N); MS *m*/*z* (%): 365 (M+, 37). Anal. Calcd: for C_18_H_15_N_5_O_2_S (365.41): C, 59.17; H, 4.14; N, 19.17. Found: C, 59.04; H, 4.05; N, 19.00%.

General procedure for the synthesis of 1,3-thiazole derivatives **6a–e**, **10a–c** and **12a–c**. 

Catalytic amounts of TEA were added into a solution of compound **3** (1 mmol, 0.365 g) and the appropriate hydrazonoyl chlorides **4a–e** or **8a–c** or α-bromoketones **11a–c** (1 mmol for each) in EtOH (20 mL), and the reaction mixture was refluxed for 4–6 h (monitored by TLC). Finally, the formed precipitate was recrystallized to give thiazoles **6a–e** or **10a–c,** or **12a–c**, respectively.

Alternate synthesis of **6a**.

A cold aqueous solution of benzenediazonium salt was added portion wise to a cold solution of compound **7** (1 mmol, 0.403 g) in pyridine (15 mL) under stirring then the precipitated product was recrystallized from DMF to give compound **6a** in 72% yield.

The physical properties and spectral data of the isolated products are listed in the Appendix A.

### 3.1. Molecular Docking

The most active compounds were sketched using Chemdraw 12.0, and their molecular modeling was performed using molecular operating environment software. The results were refined using the London DG force and force field energy. All minimizations were performed until a root mean square deviation (RMSD) gradient 0.1 kcal·mol^−1^Å^−1^ using MMFF 94× (Merck molecular force field 94×), and the partial charges were determined automatically. The binding affinity of the ligand was evaluated using the scoring function and dock function (S, Kcal/mol) created by the MOE software. The enzyme’s X-ray crystal structure (PDB ID: 4ASD, resolution: 2.03 Å) was downloaded in PDB format according to the protein data bank [51,52]. The enzyme was ready for studies using docking: (i) the water was eliminated from the protein; (ii) hydrogen atoms were added to the structure in their characteristic geometries, then reconnected the bonds broken and fixing the potential; (iii) as the large site, dummy atoms were used to execute a site search using MOE Alpha Site Finder upon this enzyme structure [53]; (iv) analyzing the ligand’s interaction with the active site’s amino acids. The best docking score is obtained as the most negative value for the active ligands. All docking procedures and scoring were recorded according to established protocols [54,55,56]. Triangle Matcher placement method and London dG score tool were used for docking.

### 3.2. Cytotoxic Assay

The cytotoxicity of the investigated substances was assessed and determined by MTT assay, and the detailed cytotoxicity assay is included in the Appendix A [57,58]. 

## 4. Conclusions

This study disclosed the design and synthesis of novel 3-thiazolhydrazinyl coumarins utilizing 3-acetyl-6-methyl-2*H*-chromen-2-one. Spectroscopy and elemental analyses were utilized to confirm the hypothesized product’s structures. Moreover, a molecular docking study of synthesized 2*H*-chromen-2-one derivatives was performed to investigate their interactions with VEGFR-2’s active region. In addition, the most active derivatives of the designed compounds were tested in vitro against the MCF-7 and LLC-Mk2 cell lines using MTT assay and Sorafenib as a reference drug. The results demonstrated the potential anti-tumor activities of compounds **6d** and **6b** (IC_50_ = 10.5 ± 0.71 and 11.2 ± 0.80 μM, respectively). Therefore, the present study demonstrated that the reported thiazolyl coumarins are potential (VEGFR-2) inhibitors and pave the way for the synthesis of additional libraries based on the reported scaffold, which could eventually lead to the development of efficient treatment for breast cancer

## Data Availability

The data presented in this study are available on request.

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
