# Peer review of "Synthesis and Molecular Docking of Some Novel 3-Thiazolyl-Coumarins as Inhibitors of VEGFR-2 Kinase"

_molecules, 2023, doi:10.3390/molecules28020689_

Round 1

Reviewer 1 Report

The manuscript “Synthesis and Molecular docking of Some Novel 3-thiazolyl-Coumarins as Inhibitors of VEGFR-2 kinase”  reports the design and in vitro assays of novel 3-thiazolhydrazinylcoumarin derivatives as VEGFR-2 inhibitors.

The manuscript is interesting and has its merits, but the authors must add some clarifications to strengthen the paper.

Major:

-- According to Protein Data Bank,  4ASD corresponds to the crystal structure of VEGFR2 in complex with sorafenib (https://www.rcsb.org/structure/4ASD); the authors must explain how they performed the molecular docking with their new ligands, as they state in Experimental that "The enzyme was ready for studies using dock-ing"

-- The authors must explain why they performed molecular docking of their new potential VEGFR ligands compared to sorafenib and then performed MTT assays using cisplatin as a reference.

-- Results and discussion, Molecular docking – The authors report that most of their compounds form H bonds with Lys868 in the VEGFR-2 active site. In this context, the authors must better describe the VEGFR-2 residues and types of bonds with sorafenib. Moreover, they must provide a more detailed description of the simulated receptor-ligand interaction in the caption of Figure 1.

Minor:

-- Introduction, the first paragraph – The authors state that “The C-terminal and N-terminal lobes each contributed residues to the region that actively binds adenosine triphosphate (ATP), which is placed in the gap between the two lobes [7].”  What protein are the authors writing about in this phrase?

-- Introduction, the second paragraph – The authors must present some examples of developed coumarin-based anticancer drugs, as well as thiazole drugs with the indicated effects, i.e., anticancer, antimicrobial, and anti-inflammatory.

-- Introduction, the third paragraph: “Molecular docking studies on VEGFR-2 crystallinity […]” – The authors must rephrase because they did not apply molecular docking to study the VEGFR's crystallinity.

-- Results and discussion, Molecular docking – The 3D representations in Figure 2 are difficult to read.

Author Response

Response to Reviewer 1 Comments

Dear reviewer, many thanks for your time spending and efforts to review the manuscript. 

Major:

Point 1: According to Protein Data Bank, 4ASD corresponds to the crystal structure of VEGFR2 in complex with sorafenib (https://www.rcsb.org/structure/4ASD); the authors must explain how they performed the molecular docking with their new ligands, as they state in Experimental that "The enzyme was ready for studies using docking"

Response 1: Thank you for your comment.

(i) The waters were eliminated from the protein. (ii) Hydrogen atoms were added to the structure in their characteristic geometries then reconnect the bonds broken and fixing the potential. (iii) As being the large site dummy atoms were used to execute a site search using MOE Alpha Site Finder upon this enzyme structure [56]. (iv) Analyzing the ligand's interaction with the active site's amino acids. The best docking Score is obtained as the most negative value for the active ligands. All docking procedures and scoring were recorded according to established protocols [57-59]. Triangle Matcher placement method and London dG score tool was used for docking.

  1. Labute, P. Protonate3D: Assignment of Ionization States and Hydrogen Coordinates to Macromolecular Structures. Proteins 2008, 75, 187–205.
  2. Kattan, S. W.; Nafie, M. S.; Elmgeed, G. A.; Alelwani, W.; Badar, M.; Tantawy, M. A. Molecular docking, anti-proliferative activity and induction of apoptosis in human liver cancer cells treated with androstane derivatives: Implication of PI3K/AKT/mTOR pathway. J. Steroid Biochem. Mol. Biol. 2020, 198, 105604.
  3. Tantawy, M. A.; Sroor, F. M.; Mohamed, M. F.; El-Naggar, M. E.; Saleh, F. M.; Hassaneen, H. M.; Abdelhamid, I. A. Molec-ular Docking Study, Cytotoxicity, Cell Cycle Arrest and Apoptotic Induction of Novel Chalcones Incorporating Thiadiazol-yl Isoquinoline in Cervical Cancer. Anti-Cancer Agents. Med. Chem. 2020, 20, 70–83.
  4. Nafie, M. S.; Tantawy, M. A.; Elmgeed, G. A. Screening of different drug design tools to predict the mode of action of ste-roidal derivatives as anti-cancer agents. Steroids 2019, 152, 108485.

Point 2:  The authors must explain why they performed molecular docking of their new potential VEGFR ligands compared to sorafenib and then performed MTT assays using cisplatin as a reference.

Response 2: Thank you for your comment.

They performed molecular docking of their new potential VEGFR ligands compared to sorafenib because: (i) Sorafenib in the VEGFR-2 as its ligand was downloaded in PDB format according to the protein data bank [54, 55]. (ii) Sorafenib is the most important, essential and common reference drug for CAS identification 284461-73-0 which 4ASD belong to this CAS identification [1].

  1. McTigue, M. Murray, B.W. Chen, J.H. Deng, Y. Solowiej, J. Kania, R.S. Molecular Conformations, Interactions, and Proper-ties Associated with Drug Efficiency and Clinical Performance Among Vegfr Tk Inhibitors molecular conformations, inter-actions, and properties associated with drug efficiency and clinical performance among Vegfr Tk Inhibitors. Proc. Natl Acad Sci USA 2012, 109-18281.
  2. Othman, I.M.M.; Alamshany, Z.M.; Tashkandi, N.Y.; Gad-Elkareem, M.A.; Anwar, M.M.; Nossier, E.S. New pyrimidine and pyrazole-based compounds as potential EGFR inhibitors: Synthesis, anticancer, antimicrobial evaluation and computa-tional studies. Bioorg. Chem. 2021, 114,105078.

[1] Anastassiadis, T.; Deacon, S. W.; Devarajan, K.; Ma, H.; Peterson, J. R. Comprehensive assay of kinase catalytic activity reveals features of kinase inhibitor selectivity. Nat. Biotechnol. 2011, 29 (11), 1039-1045.

Point 3: Results and discussion, Molecular docking – The authors report that most of their compounds form H bonds with Lys868 in the VEGFR-2 active site. In this context, the authors must better describe the VEGFR-2 residues and types of bonds with sorafenib. Moreover, they must provide a more detailed description of the simulated receptor-ligand interaction in the caption of Figure 1.

Response 3: Thank you for your comment.

Both compounds 6a and 6c form H-bond acceptor with the Lys868. It is evident that urea linker of Sorafenib plays a significant role on its binding ability for the enzyme VEGFR2 as it fits into the enzyme's allosteric site to form four significant hydrogen bonds with three essential residues (one H-bond acceptor between oxygen of urea moiety and Cys1045, one H-bond acceptor between oxygen of urea moiety and Asp1046, and finally two H-bonds donors between two NH of urea side chain and Glu885). The hydrophobic 4-chloro-3-trifluoromethylphenyl is directed toward the hydrophobic back pocket by the urea linker's binding mechanism. In addition, two hydrogen bonds; one H-bond acceptor between N atom of pyridine with essential amino acid Cys919, and another H-bond donor between the NH of the side chain with Cys919.

Minor:

Point 4: Introduction, the first paragraph – The authors state that “The C-terminal and N-terminal lobes each contributed residues to the region that actively binds adenosine triphosphate (ATP), which is placed in the gap between the two lobes [7].”  What protein are the authors writing about in this phrase?

Response 4: Thank you for your comment.

The VEGFR‐2 of protein kinases. The C-terminal and N-terminal lobes each contributed residues to the region of the VEGFR‐2 of protein kinases, which that actively binds adenosine triphosphate (ATP), which is placed in the gap between the two lobes [7].

Point 5: Introduction, the second paragraph –The authors must present some examples of developed coumarin-based anticancer drugs, as well as thiazole drugs with the indicated effects, i.e., anticancer, antimicrobial, and anti-inflammatory.

Response 5: Thank you for your comment.

We added the following paragraph: According to reports in the literature, compounds comprising coumarins, thiazoles, or thiazolylcoumarins have drawn a lot of attention from drug research due to their potential anticancer action with good IC50 [11, 12] (Fig.1).

Also, Fig. 1 was inserted into introduction to show examples of some reported coumarins, thiazoles and thiazolylcoumarins as anticancer agents:

  1. Rawat, A.; Reddy, A. V. B. Recent advances on anticancer activity of coumarin derivative. J. Med. Chem. Reports 2022, 5, 100038.
  2. Morigi, R.; Locatelli, A.; Leoni, A.; Rambaldi, M. Recent patents on thiazole derivatives endowed with antitumor activity. Recent Pat. Anticancer Drug Discov. 2015, 10, 280-297. Doi: 10.2174/1574892810666150708110432

Figure 1. Examples of some reported coumarins, thiazoles and thiazolylcoumarins as anticancer agents.

Point 6: Introduction, the third paragraph: “Molecular docking studies on VEGFR-2 crystallinity […]” – The authors must rephrase because they did not apply molecular docking to study the VEGFR's crystallinity.

Response 6: Thank you for your comment.

The molecular docking studies of these compounds were performed to confirm their ability to satisfy for the pharmacophoric features. Moreover, it also determines their binding mode interaction that occurred with the desired VEGFR‐2 inhibition.

Point 7: Results and discussion, Molecular docking- The 3D representations in Figure 2 are difficult to read.

Response 7: Thank you for your comment.

As illustrated in Fig. 3.

Reviewer 2 Report

Please, see the attachment belowe

Author Response

Response to Reviewer 2 Comments

Dear reviewer, many thanks for your time spending and efforts to review the manuscript. 

Abolibda TZ and colleagues in the article titled: Synthesis and molecular docking of some novel 3-thiazolyl-coumarins as inhibitors of VEGFR-2 kinase raised an important topic regarding the treatment of breast cancer that is still incurable (in many cases) and one of the most insidious tumors. As the efficacy of the current treatment of the breast cancer is still poor the is an urgent need to search the novel compounds that could sufficiently inhibit the proliferation and invasiveness of the breast cancer cells. It was also found the VEGR-2 is a key player in the development of the breast cancer. Moreover, overexpression of VEGFR-2

Response: Many thanks for your precious comment.

VEGFR-2 kinase exhibits conformational flexibility concerning the interaction of the inhibitors at the binding site of the receptor and significant contributor to in vivo pharmacological activity, mainly targeting cancer and other angiogenesis-associated diseases. Type II inhibitors that bind at the ‘DFG-out’ confirmation proven to be advantageous in terms of selectivity and off-target activity (side effects). The results revealed that compounds 6d and 6b exhibited greater anticancer activities (IC50 = 10.5±0.71 and 11.2±0.80 μM, respectively) than Cisplatin reference drug (IC50 = 13.3 ± 0.61 μM). Therefore, the present study demonstrated that thiazolyl coumarins are potential (VEGFR-2) inhibitors and paves the way for the synthesis of additional libraries based on the reported scaffold which could eventually lead to the development of efficient treatment for breast cancer. The most potent compound (6d) for breast cancer binds with Asp1046 amino acid (H-bond acceptor) in the DFG motif region. Compound 6b occupied HYD-II region by H-bond acceptor between nitrogen atom of thiazole moiety and Asp1046, which have potent biological activity in the micromolar (μM) range. Doi.org/10.1080/07391102.2021.1872417.

Reviewer 3 Report

1. Please add the line numbers for the current manuscript.

2. You’re developing VEGFR-2 inhibitors, why not use Sorafenib as the positive control?

3. You indicated that ‘Improved kinase selectivity and high cellular potency are just two benefits of type II inhibitors [9]’. What are the other advantages?

4. You indicated that ‘In light of our previous work on the synthesis of novel antitumor heterocycles [39–49]’. Please only list the closely related references.

5. In section 2.1. Chemistry, you can put the NMR characterization into the supporting information.

6. In section 2.2. Molecular docking section, please indicate the PDB ID for VEGFR-2. In addition, you have mentioned that the compounds have a similar binding mode as Sorafenib and therefore are potential Type II VEGFR-2 inhibitors. Please also carry out the docking study by using active VEGFR-2 with DFG-in motif for side-by-side comparison with the docking results in inactive VEGFR-2.

7. Add graphic symbols regarding the interactions for 2D binding figures. In addition, please only keep the key residues in 2D figures, and for 3D figures, only keep the residues and remove the ribbons to make the interactions clear.  

8. In the 2.3 Cytotoxic Potential section, there are many mistakes, the compound IDs do not match the content, like 8b, 8c, and 8d. Table 2 only indicates the cytotoxic activity of 6b, 6c, 6d, 10a, and 10c, however, the SAR analysis also includes other compounds. If you have tested all compounds, please list the activity results of all compounds in Table 2.

9. In the supporting information, many MS results do not match the corresponding molecular weights.

10. Please revise the current manuscript thoroughly, there are many errors and typos. Please double-check and make sure the information you provided is correct.

Author Response

Response to Reviewer 3 Comments

Dear reviewer, many thanks for your time spending and efforts to review the manuscript. 

  1. Point 1: Please add the line numbers for the current manuscript.

Response 1: Done (see revised manuscript)

  1. Point 2: You’re developing VEGFR-2 inhibitors, why not use Sorafenib as the positive control?

Response 2: Thank you for your comment.

Because; (i) we used Cisplatin. (ii) As we conduct biological activity on cancer cell lines, Cisplatin was utilized as a reference compound.

  1. Point 3: You indicated that ‘Improved kinase selectivity and high cellular potency are just two benefits of type II inhibitors [9]’. What are the other advantages?

Response 3: Thank you for your comment.

Type II inhibitors stabilize an inactive VEGFR‐2 kinase form that features the DFG motif in a DFG‐out conformation, creating an allosteric hydrophobic pocket directly adjacent to the ATP‐binding site.  

It represented in publicly available biochemical profiling studies of kinase inhibitors is very small. It
is likely that the relative contribution to the binding affinity of the reorganization free energy change associated with the DFG-in to DFG-out transition is different for different kinases. The overall importance of the DFG-in to DFG-out reorganization free energy compared with the binding energy in the selectivity of inhibitors for individual kinases remains to be determined.  Doi.org/10.1021/jm501603h.

The essential pharmacophoric features shared by sorafenib and most of VEGFR‐2 type II inhibitors[2–4] are as follows: (a) a flat heterocyclic core such as thiazole moiety that occupies the ATP‐binding region and orients an essential H‐bond acceptor (a lone pair of nitrogen) to the backbone donor (of Cys919 residue); (b) a central aromatic ring (hydrophobic spacer), which occupies the linker region between the ATP‐binding domain and the DFG domain; (c) urea or amide moiety, acting as H‐bond donor–acceptor pair, aiming interaction with Glu885 and Asp1046 residues in the DFG‐out domain (allosteric site); the NH motif forms a hydrogen bond with Glu885, and the CO motif forms a hydrogen bond with Asp1046; (d) the terminal aryl moiety, another feature, occupies the allosteric hydrophobic back pocket, making hydrophobic interactions with the hydrophobic side chains of Ile888, Ile892, Leu1019, and Ile1049 [5].

Considering all the above facts, developing the more selective and competitive type II small molecule
inhibitors is an effective therapeutic approach.

[2] A. A. El‐Helby, H. Sakr, I. H. Eissa, H. Abulkhair, K. El‐Adl, Arch. Pharm. Chem. Life Sci. 2019, 352, 1900113.
[3] A. A. El‐Helby, H. Sakr, I. H. Eissa, A. A. Al‐Karmalawy, K. El‐Adl, Arch. Pharm. Chem. Life Sci. 2019, 352, 1900178.
[4] H. A. Mahdy, M. K. Ibrahim, A. M. Metwaly, A. Belal, A. B. M. Mehany, K. M. A. El‐Gamal, A. El‐Sharkawy, M. A. Elhendawy, M. M. Radwan, M. A. Elsohly, I. H. Eissaa, Bioorg. Chem. 2020, 94, 103422.
[5] Q. Q. Xie, H. Z. Xie, J. X. Ren, L. L. Li, S. Y. Yang, J. Mol. Graphics Modell. 2009, 27, 751.

  1. Point 4: You indicated that ‘In light of our previous work on the synthesis of novel antitumor heterocycles [39–49]’. Please only list the closely related references.

Response 4: Done

  1. Point 5: In section 2.1. Chemistry, you can put the NMR characterization into the supporting information.

Response 5: Thank you for your comment.

Authors think that the discussion of spectral data of the prepared new compounds in the main manuscript clarifies the reactions

  1. Point 6: In section 2.2. Molecular docking section, please indicate the PDB ID for VEGFR-2. In addition, you have mentioned that the compounds have a similar binding mode as Sorafenib and therefore are potential Type II VEGFR-2 inhibitors. Please also carry out the docking study by using active VEGFR-2 with DFG-in motif for side-by-side comparison with the docking results in inactive VEGFR-2.

Response 6: Thank you for your comment.

Compound

Energy score (S)
(kcal/mol) of type II VEGFR-2 inhibitor with DFG-out

Energy score (S)
(kcal/mol) of type I VEGFR-2 inhibitor with DFG-in

3

-7.723

-7.319

6a

-9.156

-9.155

6b

-9.819

-9.089

6c

-9.583

-9.580

6d

-9.900

-9.857

6e

-9.377

-9.254

10a

-9.673

-9.205

10b

-9.421

-8.728

10c

-9.690

-9.572

12a

-8.523

-8.135

12b

-8.901

-8.363

12c

-8.959

-8.735

Sorafenib

-9.284

-8.214

  1. Point 7: Add graphic symbols regarding the interactions for 2D binding figures. In addition, please only keep the key residues in 2D figures, and for 3D figures, only keep the residues and remove the ribbons to make the interactions clear.  

Response 7: Thank you for your comment.

Illustrating the abbreviations of amino acid residues; Asp: Aspartic acid (D), Phe: Phenylalanine (F), Glu: Glycine (G), Lys: Lysine (K), Leu: Leucine (L), Arg: Arginine (R), Cys: Cysteine (C), Gly: Glutamic acid (E), Ser: Serine (S), and Ile: Ile: Isoleucine (I). DOI: 10.1016/s0008-6215(97)83449-0.

As illustrated in figure 2.

  1. Point 8: In the 2.3 Cytotoxic Potential section, there are many mistakes, the compound IDs do not match the content, like 8b, 8c, and 8d. Table 2 only indicates the cytotoxic activity of 6b, 6c, 6d, 10a, and 10c, however, the SAR analysis also includes other compounds. If you have tested all compounds, please list the activity results of all compounds in Table 2.

Response 8: Compounds 8b, 8, 8d were corrected to 6b, 6c, 6d

The SAR based on experimental measurements includes only the in vitro cytotoxic activity of thiazoles 6b, 6c, 6d, 10a and 10c against MCF-7, and LLC-MK2.  While the SAR based on Molecular docking calculations includes all the prepared compounds

  1. Point 9: In the supporting information, many MS results do not match the corresponding molecular weights.

Response 9: We were checked all mass with their molecular weights and the mass of compounds 12b and 12c

  1. Point 10: Please revise the current manuscript thoroughly; there are many errors and typos. Please double-check and make sure the information you provided is correct.

Response 10: The whole manuscript was revised for typographical and grammatical errors

Round 2

Reviewer 1 Report

The authors inserted the observations corresponding to the reviewer's comments in the revised manuscript without indicating the page number or the section/subsection, thus making it difficult to follow the correlation between the response letter and the revised manuscript. As a friendly note, the authors should add at the end of each response an indication about the page/lines/section where they introduced new text.

Point 1. The authors added detailed explanations on how they prepared the enzyme for the molecular docking. As such, the expression “the enzyme was ready for studies using docking” is inappropriate. The authors should rephrase/replace this expression (for example, see the Docking protocol in https://www.mdpi.com/1999-4923/12/9/879).

 Point 2. The authors have yet to respond to the second part of my comment. Indeed, they correctly argued the use of sorafenib for molecular docking but did not argue why they chose cisplatin as a reference in their in vitro tests.

Author Response

Response to Reviewer 1 Comments

Dear reviewer, many thanks for your time spending and efforts to review the manuscript. 

As a friendly note, the authors should add at the end of each response an indication about the page/lines/section where they introduced new text.

Response 1: Done

Point 1. The authors added detailed explanations on how they prepared the enzyme for the molecular docking. As such, the expression “the enzyme was ready for studies using docking” is inappropriate. The authors should rephrase/replace this expression (for example, see the Docking protocol in https://www.mdpi.com/1999-4923/12/9/879).

Response 1:  

 In the docking protocol, for the protein target, we followed these reported steps: (i) The waters were eliminated from the protein. (ii) Hydrogen atoms were added to the structure in their characteristic geometries then reconnect the bonds broken and fixing the potential. (iii) As being the large site dummy atoms were used to execute a site search using MOE Alpha Site Finder upon this enzyme structure [53]. (iv) Analyzing the ligand's interaction with the active site's amino acids. The best docking Score is obtained as the most negative value for the active ligands. All docking procedures and scoring were recorded according to established protocols [54-56]. For the docking process, we chose the triangle Matcher placement method and London dG score tool. Page 15/ Lines 14, 15, 22, and 23.

 Point 2. The authors have yet to respond to the second part of my comment. Indeed, they correctly argued the use of sorafenib for molecular docking but did not argue why they chose cisplatin as a reference in their in vitro tests.

Response 2:  

We used sorafenib as a reference drug for the in vitro examination against both human breast cancer (MCF-7) and normal cell line LLC-Mk2 cell lines with IC50 values of 5.10 ± 0.49 and 135.3±4.08 μM, respectively.

Reviewer 3 Report

The current manuscript is revised based on the reviewers' comments. My recommendation is 'Accept in present form'.

Author Response

Dear reviewer, many thanks for your time spending and efforts to review the manuscript.